# Synthetic chimeric nucleases function for efficient genome editing

R.M. Liu[1], L.L. Liang[1], E. Freed[1], H. Chang[2], E. Oh[1], Z.Y. Liu[2], A. Garst[3], C.A. Eckert [1,4] & R.T. Gill[1,5]*

CRISPR–Cas systems have revolutionized genome editing across a broad range of biotechnological endeavors. Many CRISPR-Cas nucleases have been identified and engineered for improved capabilities. Given the modular structure of such enzymes, we hypothesized that engineering chimeric sequences would generate non-natural variants that span the kinetic parameter landscape, and thus provide for the rapid selection of nucleases fit for a particular editing system. Here, we design a chimeric Cas12a-type library with approximately 560 synthetic chimeras, and select several functional variants. We demonstrate that certain nuclease domains can be recombined across distantly related nuclease templates to produce variants that function in bacteria, yeast, and human cell lines. We further characterize selected chimeric nucleases and find that they have different protospacer adjacent motif (PAM) preferences and the M44 chimera has higher specificity relative to wild-type (WT) sequences. This demonstration opens up the possibility of generating nuclease sequences with implications across biotechnology.

[1] Renewable and Sustainable Energy Institute (RASEI), University of Colorado, Boulder, CO, USA. [2] Department of Biochemistry, University of Colorado, Boulder, CO, USA. [3] Inscripta, Inc., Boulder, CO, USA. [4] National Renewable Energy Laboratory, Golden, CO, USA. [5] NNF-Center for Biosustainability, Danish Technical University, Lyngby, Denmark. *email: rtg@biosustain.dtu.dk

CRISPR-Cas (clustered regularly interspaced short palindromic repeats) driven genome editing and engineering has dramatically impacted biology and biotechnology in general[1,2]. CRISPR-Cas editing systems require a polynucleotide guided nuclease, a guide polynucleotide (i.e., a guide RNA (gRNA)) that directs by homology the nuclease to cut a specific region of the genome, and optionally, a donor DNA cassette that can be used to repair the cut dsDNA and thereby incorporate programmable edits at the site of interest. The earliest demonstrations and applications of CRISPR-Cas editing used Cas9 nucleases and associated gRNA[3–6]. These systems have been used for gene editing in a broad range of species encompassing bacteria to higher order mammalian systems[4,5,7–13]. It is well established, however, that key editing parameters such as protospacer adjacent motif (PAM) specificity, editing efficiency, and off-target rates, among others, are species, loci, and nuclease dependent. As such, there is intense interest in identifying and rapidly characterizing nuclease systems that can be exploited to broaden and improve overall editing capabilities[2,14].

To this end, Cas12a-type nucleases have emerged as suitable alternatives to Cas9 nucleases, where several nucleases (e.g., *Acidaminococcus* sp. (AsCas12a) and *Lachnospiraceae bacterium* (LbCas12a)) have now been shown to display comparable genome-editing capability to Cas9 while providing different PAM preferences[15–19]. We employ the term "Cas12a-type" or "-like" throughout in recognition of consistent changes in CRISPR-Cas evolutionary classification and naming schemes[14,20]. The structures of AsC12a/LbCas12a contain a bi-lobed architecture consisting of an α-helical recognition (REC) lobe and a nuclease (NUC) lobe with a positively charged channel between them that binds the crRNA-DNA hybrid[15,21] (Fig. 1a). The REC lobe consists of the REC1 and REC2 domains, and the NUC lobe consists of the RuvC domain and three additional domains, which are referred to as the WED, PI, and Nuc domains, respectively[15,21] (Fig. 1a). The WED domain is assembled from three regions (WED-I, WED-II, and WED-III) in the Cas12a sequence (Fig. 1a). The REC lobe (REC1 and REC2) is located between the WED-I and WED-II regions, and the PI domain is inserted between the WED-II and WED-III regions (Fig. 1a). The RuvC domain contains the three motifs (RuvC-I, RuvC-II, and RuvC-III) (Fig. 1a). The bridge helix (BH) is located between the RuvC-I and RuvC-II motifs and connects the REC and NUC lobes, whereas the Nuc domain is inserted between the RuvC-II and RuvC-III motifs (Fig. 1a).

The canonical Cas12a protein–RNA complex recognizes a T-rich PAM and leads to a staggered DNA double-stranded break[19,22] (Fig. 1a). The Cas12a-like nuclease interacts with the pseudoknot structure formed by the 5′-handle of crRNA[22,23]. A guide RNA segment, composed of a seed region and the 3′ terminus, possesses complementary binding sequences with the target DNA sequences. Cas12a-like nucleases characterized to date have been shown to work with a single gRNA and to process gRNA arrays[21]. In addition, Kim et al. compared the ratio of total off-target with on-target modification for AsCas12a and LbCas12a, and found that both orthologs show lower off-target activity than that previously observed with SpCas9[24]. While Cas12a and Cas9 like nuclease systems have proven highly impactful, neither system has been shown to function as predictably as is desired to enable the full range of applications envisioned[2,14].

A range of efforts have attempted to engineer improved CRISPR editing systems, which included engineering of the PAM specificity, stability, and sequence of the gRNA and-or the nuclease[25–32]. For example, chemical modifications of CRISPR–Cas9 gRNA expected to increase gRNA stability did indeed lead to 3.8-fold higher indel frequencies in human cells[27].

In addition, Gao et al. performed structure-guided mutagenesis of Cas12a and screened to identify variants with an increased range of recognized PAM sequences. The engineered AsCas12a recognized TYCV and TATV PAMs in addition to the canonical TTTV, with enhanced activities in vitro and in human cells[29]. Using the crystal structures of Cas9[22,33], Slaymaker et al. performed rational engineering of the DNA binding region to attempt to decrease binding under the hypothesis that this would result in a lower off-target editing rates[25]. Indeed, the authors reported eSpCas9(1.0) and eSpCas9(1.1) mutants that decreased 50% cleavage with off-target sites (<0.2% indel) relative to wild-type (WT) Cas9 using 20-nucleotide RNA guides[25]. Strohlkendl et al. reported that Cas12a binding to target DNA is rate limiting for cleavage, and suggested ways to engineer nucleases focused on altering such binding ($k$on)[19].

Here, we demonstrate a platform for construction of a library of synthetic nucleases that span the kinetic space encompassing overall editing considerations. In our design process we specifically note that even though Cas12a-like nucleases exhibit considerable overall sequence diversity, they also retain several conserved regions that we hypothesize may be recombined in a modular fashion from one Cas12a template to another to produce active chimeric nucleases. We hypothesize that such nucleases would exhibit altered kinetic characteristics that would allow for the discovery of sequences with desired editing characteristics. To test this hypothesis, we design and build, in *Escherichia coli*, a Cas12a-type nuclease library with 560 mutants that combined up to six conserved regions from a diverse starting pool of Cas12a-like nucleases (Supplementary Fig. 1 and Supplementary Table 1). We then select and screen for functional Cas12a-type chimeras, identifying several dozen that show basal editing. To then demonstrate the use of this strategy as a platform for rapid generation of nucleases with altered characteristics, we characterize several of the most active chimeras and demonstrate altered PAM preferences and on- vs off-targeting. The best performing chimera, M44, has higher specificity than all other tested nucleases. Finally, we demonstrate that such nucleases retain activity in yeast (*Saccharomyces cerevisiae*) and human cells (HEK293T), thus opening up the use of this strategy to rapidly build and select for synthetic nucleases with desired characteristics across a broad range of applications.

## Results

**Structure-guided design of Cas12a-type chimera library**. A broad genetic and functional diversity of CRISPR-Cas immune systems have been identified across bacteria and archaea[14]. Here, we sought to demonstrate a platform for rapid generation of a diverse set of nuclease sequences that would expand desired activities for genome-editing applications. We reasoned that instead of relying on isolation and characterization of Cas12a-like nucleases, the generation of chimeric sequences from known orthologs could more effectively expand the search space for the selection of novel functions[34,35]. In addition, the module-like nature of CRISPR nucleases would facilitate domain recombination strategies for chimera generation[36]. To this end, we selected nine Cas12a-type gene sequences (Supplementary Fig. 1 and Supplementary Table 1) as our initial starting pool that spanned a broad sequence space (560 variants) and tested their editing efficiency in *E. coli* (Fig. 1b). All of the tested nucleases were functional in *E. coli* using the *galK* inactivation assay[7,37], and the transformation efficiencies (CFU/μg) of all the nucleases were at or higher than 10^5 (Fig. 1b).

We employed a structure-guided design approach to build our chimeric nuclease library. We first aligned our nine Cas12a-like nucleases' sequences with the AsCas12a and LbCas12a sequences

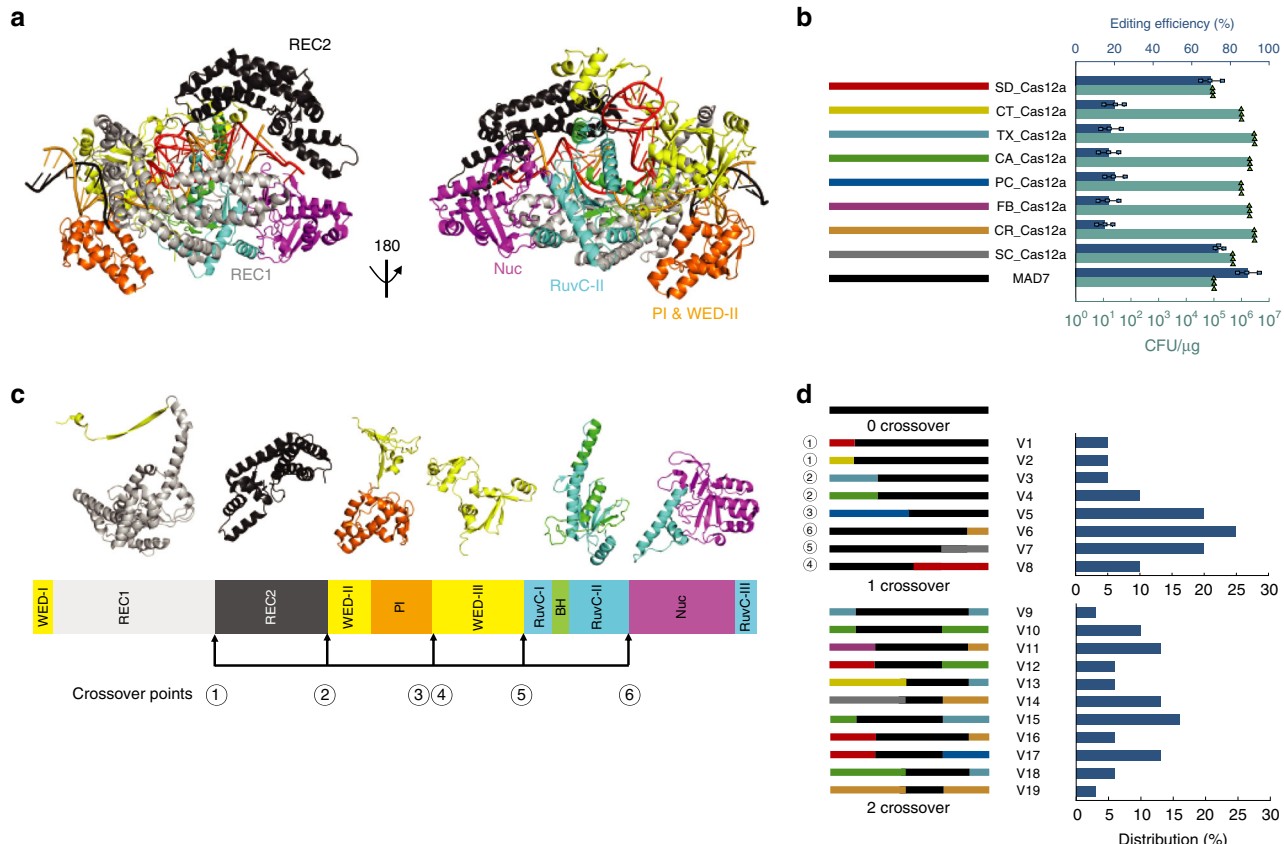

**Fig. 1** Cas12a-type chimera library construction. **a** The Cas12a-like protein structure analysis based on AsCas12a (PDB:5B43). **b** The editing and transformation efficiencies of the Cas12a-like nucleases used in this study. The Cas12a-like nucleases, used in this study, are SD_Cas12a (*Succinivibrio dextrinosolvens*), CT_Cas12a (*Candidatus Methanoplasma termitum*), TX_Cas12a (*Thiomicrospira sp.* XS5), CA_Cas12a (*Candidatus Methanomethylophilus alvus*), PC_Cas12a (*Porphyromonas crevioricanis*), FB_Cas12a (*Flavobacterium branchiophilum*), CR_Cas12a (*Candidatus Roizmanbacteria bacterium* GW2011_GWA2_37_7), SC_Cas12a (synthetic construct of AsCas12a), and MAD7. The editing efficiency, as determined by *galK* inactivation assay, was shown in blue, and the transformation efficiency was shown in green. **c** The domains were separated by the six crossover points, which were WED-I&REC1, REC2, WED-II&PI, WED-III, RuvC-1&BH&RUVC-II, and Nuc&RuvC-III. **d** The distribution of the chimera library variants. Cas12a-type libraries were made as described in the online methods. The numbers to the left of variants were the crossover points shown in (**c**). The swapped regions are shown by different colors; the colors are the same as in (**b**). Source Data are available in the Source Data File.

(Supplementary Figs. 2 and 3) and observed a relatively low level (33.8–42.99%) of sequence identity (Supplementary Table 2), thus limiting homology driven strategies and constraining the mutational space we could explore via chimeragenesis. Therefore, we had to identify modules based on known structures from AsC12a/LbCas12a where we could design junction points for crossover. Based on the sequence alignment results, we identified six crossover points for chimeragenesis (Fig. 1c, Supplementary Fig. 2). We chose regions that spanned at least 500 bp and contained one or more functional domains. Each crossover point had higher amino acid conservation across orthologs than the surrounding sequence (Supplementary Fig. 2). These crossover points are designed to include the following chimeric domains: (1) REC1; (2) REC1/RCE2; (3) REC1/REC2/WED-II/PI; (4) WED-III/RuvC-I/BH/RuvC-II/NuC/RuvC-III; (5) RuvC-I/BH/ RuvC-II/NuC/RuvC-III; (6) NuC/RuvC-III. The libraries were constructed in parallel (online method, Supplementary Fig. 2), and the distribution of library variants was not higher than 25 and 15% in 1 and 2 crossover library construction, respectively (Fig. 1d).

**Isolation of functional Cas12a-type chimeras.** Functional chimeras were selected using both a *galK* based growth selection[38] and *galK* colorimetric screen[7,37] (Fig. 2a and Supplementary

Fig. 4). Selected variants were then characterized by sequencing of the plasmid insert region containing the chimeric gene. We expected to observe either (1) a small number of chimeric Cas12a-type protein sequences that retain specific modules to produce a spectrum of activities, or (2) a large number of overall chimeric Cas12a-type protein sequences with a small number specific recombined domains.

Our sequencing results identified 24 chimeric sequences after selection, eight of which were identified more than once in the sequenced pool (Supplementary Table 1 and Supplementary Fig. 5). Notably, all eight chimeric Cas12a-type variants were characterized by recombination at the REC1 lobe, which has not, to our knowledge, been reported previously. We also identified one crossover at position 6 that defines the boundary between RuvC-II and the NUC domains. All of these eight exhibited reasonable editing efficiency, with two (M44, M21) exhibiting editing efficiency approaching that of the best WT nuclease sequence (MAD7 from Fig. 1b compared with Fig. 2b, c) when confirmed by the *galK* based colorimetric screen (Fig. 2a, b). The second crossover at position 6 in M22 significantly decreased the editing efficiency compared with M44, which suggested REC1 domain had potential flexibility compared with other domains. While these data provide solid evidence that this method for chimeragensis is a viable approach for the generation of non-

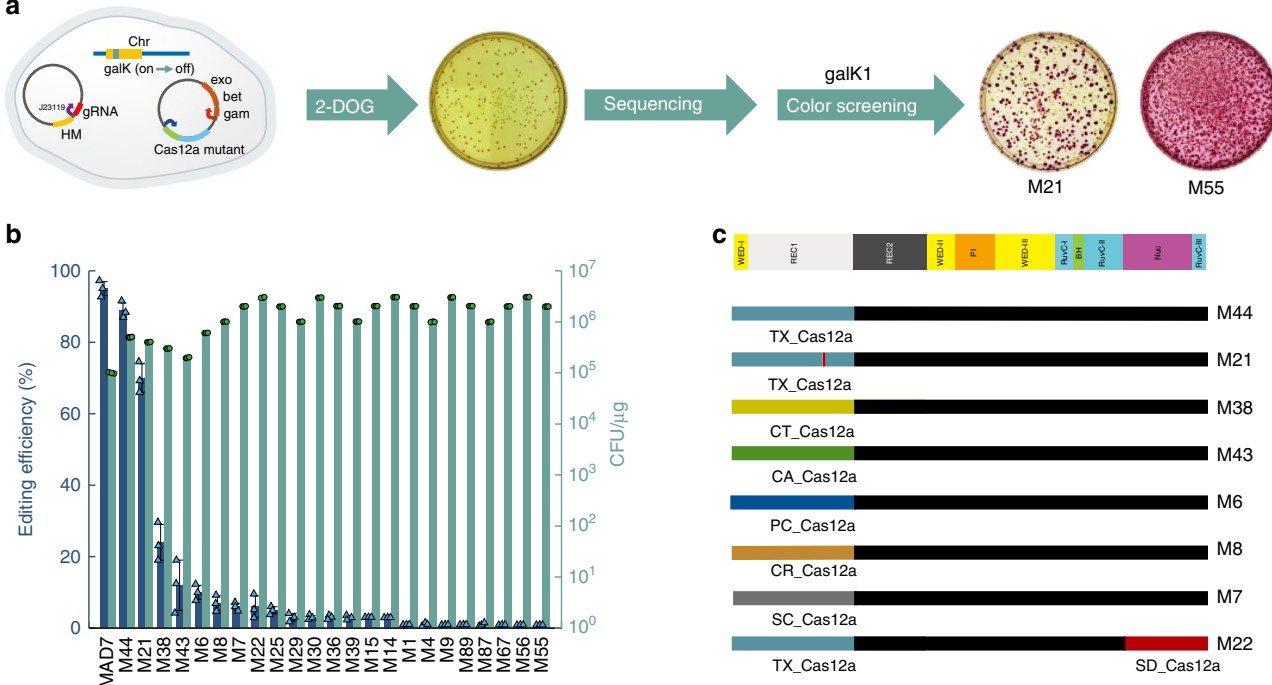

**Fig. 2** Verification of functional chimera library variants. **a** The workflow to verify positive variants. We constructed a two plasmid system for genome editing: one plasmid expresses a Cas protein as well as lambda red proteins (exo, bet, and gam)[67]; a second plasmid expresses a single crRNA (with J23119 promoter) targeting the *galK* gene and a homology arm (HM) containing a *galK*-inactivating mutation as a template for recombineering. Strains with GalK inactivation can grow in media supplemented with 2-DOG, while any unedited strains will not grow, eliminating the need for further screening. In addition, the positive mutants were evaluated for editing efficiency in *E. coli* using a color screening method based on GalK inactivation. In this assay, colonies with active GalK are red and colonies with inactive (edited) GalK are white. **b** The editing and transformation efficiency test of library variants. Transformation efficiency is defined as the number of colony forming units (cfu) per μg of gRNA plasmid. **c** The diagram of positive chimera variants based on the Cas12a structure. (The structure of AsCas12a used as the template.) Source Data are available in the Source Data File.

natural nuclease sequences, in our experience the results from a single editing assay are not sufficient to understand the functional capabilities of an editing system.

**Verification of functional Cas12a-type variants.** We further verified the activity of selected Cas12a-type chimeras using several additional editing assays and the MAD7 nuclease for comparison. We specifically selected five additional inactivating mutations positioned in the *galK* and *lacZ* genes that could be used to measure nuclease-mediated cell killing (gRNA directed cutting only) (online methods, Fig. 3a, b) and nuclease-mediated editing via color screening (Fig. 3c and Supplementary Fig. 6). Mutants M44 and M21 (which is identical to M44 except for a single G218A substitution), displayed the same cutting efficiency (100%) with WT control using four of six designed gRNAs targeted on *galK* and *laz* genes (Fig. 3b). However, not every gRNA elicits cleavage although the gRNAs were targeted in the same gene. Low activity could result from either failure to form a functional Cas12a–gRNA complex or inability to recognize targets in vivo[39–41]. Then, the four gRNAs with high cutting efficiency were used for editing constructs as a representative sample and tested the inactivation editing at different position of *galK* and *lacZ* genes. We observed that the editing efficiencies of MAD7, M44, and M21 were not as high as their cutting efficiencies. Using galK2 and lacZ1 gRNAs, the editing efficiencies of M21 and M44 were 12.5% and 38% higher than MAD7. Using galK1 and lacZ2, the editing efficiency of MAD7 was 7% and 60% higher than M44 (mutants with highest editing efficiency) (Fig. 3c). These results indicated that there were more factors affecting the recombineering using Cas12a-type chimeras. Notably, the transformation efficiency for all the Cas12a-type chimeras

were 2–5-fold higher than MAD7 (Supplementary Fig. 7). These results suggested the nuclease-mediated cell killing of Cas12a-type chimeras were not as strong as MAD7, which could be an important feature when working with species that are more difficult to transform or attempting to construct large mutant libraries.

After testing different targets at the same position on the genome, we next tested how the editing capability of chimeras changed as a function of targeting different positions in the genome. We specifically targeted five distinct "safe sites", which are non-essential sites in *E. coli* BW25113 genome chosen for integration of heterologous genes with minimal predicted side effects[42,43]. We deleted the WT *galK* gene, and then integrated the *galK* gene with a strong constitutive promoter J23119 into the five chosen safe sites (Supplementary Fig. 8a). Similar to the prior results, editing efficiency of M44, M21, and M38 was confirmed albeit with consistently lower efficiency when compared with MAD7 (Supplementary Fig. 8b, c). We also noted a positional dependency conserved across all testing nucleases, with higher editing efficiency observed closer to the origin of replication (Supplementary Fig. 8b, c).

**Increased expression increases editing efficiency.** The editing efficiency of the characterized chimeras spanned a range of 5–95% depending on the specific PAM targeted (galk, lacZ) or the specific loci in the genome targeted. This broad range was consistent with our hypothesis that chimeric sequences would not only be functional but would provide a range of kinetic capabilities that could be used as starting points in screens or selections for desired performance (e.g., on- vs off-targeting). The core supposition here is that chimeric sequences are less stable

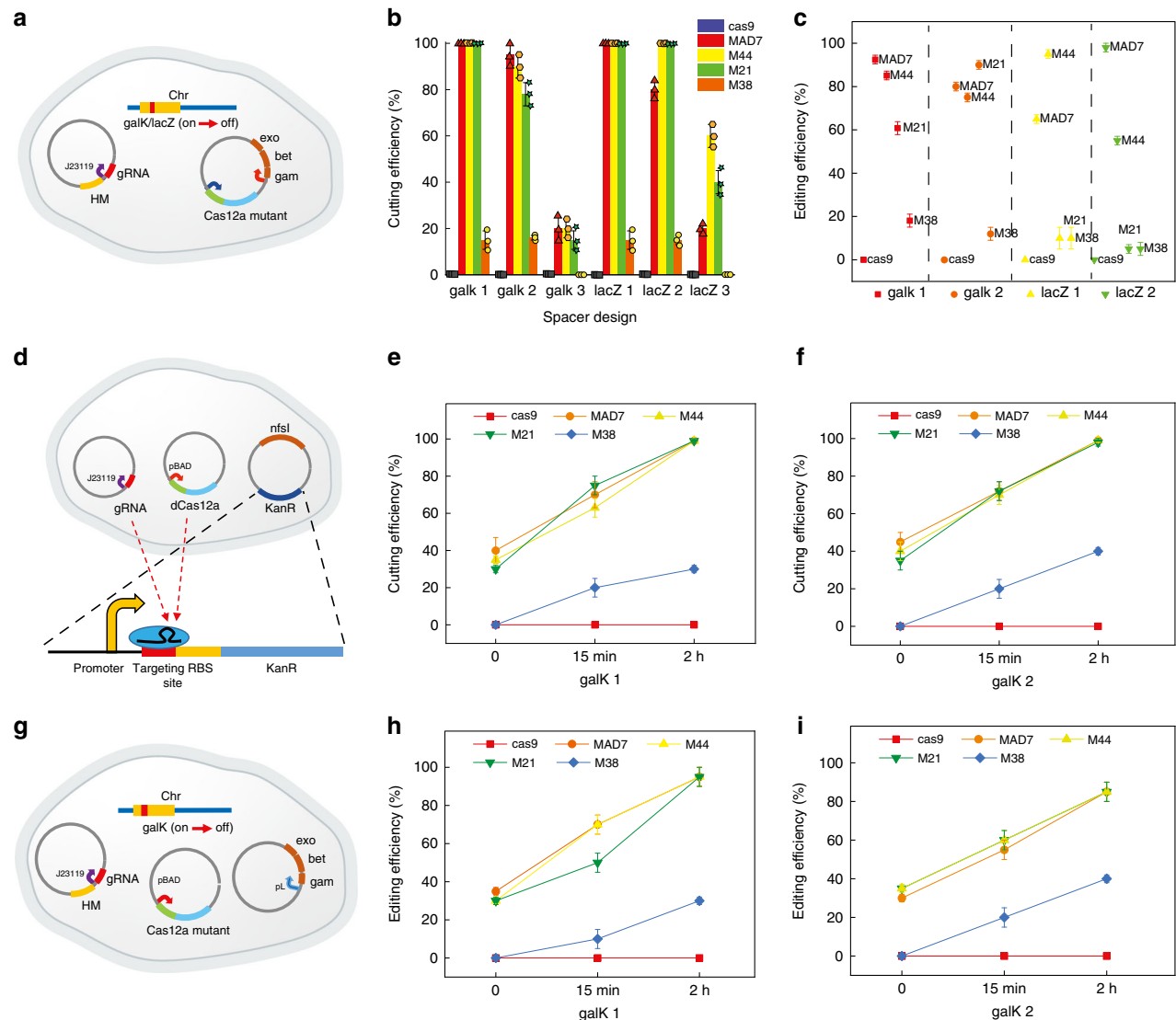

**Fig. 3** The genome-editing test with different gRNAs for chimera library variants in *E. coli*. **a** Editing (cutting) efficiency test using gRNA targeting *galK* or *lacZ* genes. We constructed a two plasmid system for genome editing: one plasmid expresses a Cas protein as well as lambda red proteins (exo, bet, and gam)[67]; a second plasmid expresses a single crRNA (with J23119 promoter) targeting the *galK* or *lacZ* gene and a homology arm (HM) containing a gene-inactivating mutation. For the cutting, there were no lambda red proteins or homology arm in the system. **b** Cutting efficiency of chimeric Cas12a-like proteins using six different gRNA plasmids (online methods). The gRNA plasmids galK1, galK2, and galK3 targeted different positions in the *galK* gene. The gRNA plasmids lacZ1, lacZ2, and lacZ3 targeted different positions in the *lacZ* gene. **c** The editing efficiency of chimera library variants with different gRNAs. The gRNAs used in the test were galK1, galK2, lacZ1, and lacZ2. Editing efficiency was determined by color screening (red/white for GalK or blue/white for LacZ). **d** The dCas12a (or Cas12a with reduced activity) protein binding assay. We constructed a three plasmid system: one plasmid expresses dCas12a (or Cas12a with reduced activity) using a arabinose inducible promoter (pBAD); a second plasmid expresses a single crRNA (with J23119 promoter) targeting the *kanR* gene; a third plasmid expresses the kanamycin resistance protein (encoded by *kanR* gene) using a constitutive promoter containing a fully complementary (on-target) crRNA binding site as well as a nitroreductase (encoded by *nfsI* gene) which makes the cells sensitive to metronidazole. **e, f** The cutting efficiency of chimeric Cas12a-like nucleases with different arabinose induction times using different gRNA. **e** galK_1 and **f** galK_2. **g** The arabinose inducible system for chimeric Cas12a-like proteins. We constructed a three plasmid system for genome editing: one plasmid expresses a Cas12a-like protein using a arabinose inducible promoter; a second plasmid expresses lambda red proteins (exo, bet, and gam) using a temperature-inducible promoter (pL); a third plasmid expresses a single crRNA (with J23119 promoter) targeting the *galK* gene with homology arm (HM) containing a *galK*-inactivating mutation as a template for recombineering. **h, i** The editing efficiency of chimeric Cas12a-like nucleases with different arabinose induction times using different gRNA. **h** galK_1 and **i** galK_2. Source Data are available in the Source Data File.

than WT sequences since they have not been selected for function in nature. Lowered stability could affect function broadly, including altered on-/off-targeting and cleavage kinetic constants, or overall concentrations due to increased degradation in vivo[44].

To investigate the above supposition, we sought to evaluate the chimeric Cas12a-like proteins through the process of CRISPR editing, included on-targeting binding, cutting, and editing

associated with recombineering proteins. Thus, we first developed a dCas12a (or Cas12a with greatly reduced activity) protein binding assay that allowed the on-target and off-target status to be monitored by antibiotic selections in *E. coli* (Fig. 3d). We constructed a three plasmid system that expresses dCas12a (or Cas12a with greatly reduced activity) using a arabinose inducible promoter, a single crRNA (with J23119 promoter) targeting *kanR*

gene, and a kanamycin resistance protein (encoded by *kanR* gene) using a constitutive promoter containing a fully complementary (on-target) crRNA binding site as well as a nitroreductase (encoded by *nfsI* gene) which conferred the cells sensitive to metronidazole[45] (Fig. 3d and Supplementary Fig. 9). We introduced two tested on-target crRNA (galK1 and galK2) binding sites in the upstream of *kanR* gene individually. Using dCas12a:crRNA as a transcriptional repressor, the cells cannot grow with unexpressed kanamycin, and expressed nitroreductase also repress the cell growth with metronidazole (Fig. 3d).

The decreased cell grown under antibiotic selection means the dCas12a:crRNA repressed the transcription of kanamycin resistance protein. However, repression levels of chimeras were 5–60% lower than the WT MAD7 nuclease (Supplementary Fig. 10). Recent publications described that DNA target binding by CRISPR-Cas12a is rate limiting for DNA cleavage, and the maximal rate constant ($k$max) for the targeted DNA binding of (AsCas12a) is $0.13 \pm 0.01 \, \mathrm{s}^{-1}$ which is orders of magnitude slower than the DNA cleavage[19]. In light of our DNA binding assay data, we hypothesized that the chimeric Cas12a-like proteins were less stable than the WT Cas12a under the same expression level, and increased chimera degradation may limit the overall DNA binding rate.

To test this hypothesis, we introduced the Cas12a-like proteins into three plasmid system to replace the dCas12a, and tested the cutting efficiency with the different expression level of Cas12a proteins by controlling the induction time. Prolonged induction increases the Cas12a expression level, which resulted in increased cutting efficiency of chimeric Cas12a-like proteins (Fig. 3e, f). In addition, we also tested the genome editing under this inducible system (Fig. 3g). The results showed that the editing efficiency of chimeric Cas12a-type mutants were further improved with longer induction time, and the editing efficiency of M44 and M21 were similar to the WT Cas12a after 2 h induction (Fig. 3h, i). These studies support the hypothesis that the studied chimeric nucleases are less stable than WT sequences and that this effect can be mitigated by increased expression. Again, these data demonstrate that chimeric sequences do indeed provide for the development of editing systems that span a broad kinetic landscape.

**Altered PAM preferences and off-target editing rates**. Prior efforts have engineered CRISPR nucleases to achieve reduced off-targeting and altered PAM preferences by targeted mutagenesis of the modules responsible for such activities[25–29]. The thought in such studies was to reduce or alter overall DNA binding such that off-target events would not be energetically favorable enough to allow for cutting. We utilized similar rationale in our chimera strategy, and thus expected that the functional chimeras identified may have altered PAM preferences and/or off-target rates.

We first tested the PAM preferences of three of the selected chimeras. To elucidate functional PAM sequences, we developed a high-throughput in vivo screen with two features: applicability across PAM-dependent CRISPR-Cas systems and the generation of a distinct signal for functional PAMs. More recent efforts have developed high-throughput experimental screens to determine functional PAMs based on the depletion of a target plasmid or on the introduction of a double-stranded break in vitro[46–48]. Here, we modified the Cas12a binding assay described above to generate a comprehensive screen to elucidate the complete landscape of functional PAM sequences (Supplementary Fig. 11a). We constructed the reporter plasmid containing *KanR* gene encoding for kanamycin resistance and the functional proto-spacer with NNNN PAM library. We then transformed the chimeric Cas12a-like proteins and two of equivalent gRNA

plasmids individually into the *E. coli* MG1655. One gRNA design is targeted on the *KanR* gene, and another gRNA is a non-targeting control. We collected the cells grown on kanamycin media using different gRNA plasmids, and amplified the region of the PAM library from the reported plasmid for the high-throughput sequencing.

The PAM enrichment score revealed that PAM preferences appeared to differ among the chimeric Cas12a-like proteins tested (Fig. 4, Supplementary Fig. 11b–h, 12). Interestingly, the TTTC PAM is still the top one for the tested chimeric Cas12a-like and WT Cas12a proteins (expect TX_Cas12a) (Fig. 4, Supplementary Fig. 12). Furthermore, we observed that CTTT PAM with the lowest enrichment score for all known PAMs (Fig. 4, Supplementary Fig. 12), which may be explained by weak and strong PAMs all eliciting irreversible DNA damage and infrequent escape[49–51]. To confirm these high-throughput observations, we tested several of the previously unreported PAMs individually, and the results also revealed PAM specificity among different chimeric Cas12a-like proteins (Supplementary Fig. 13).

Off-target mutations observed at frequencies greater than desired is still a major concern when applying CRISPR systems to biomedical and clinical application[24,48,52,53]. Several prior studies have engineered altered off-target rates by site-directed and random mutagenesis on CRISPR nucleases to decrease non-specific interactions with target DNA[25,26,54,55]. Thus, we expected that our chimera strategy may affect the nonspecific interactions with the target DNA site. We carried out an off-targeting library test in which we assessed the effects of systematically mismatching various positions within gRNAs. We designed nine of such off-target cassettes, including 3 each with substitutions, insertions, or deletions in different positions (Fig. 5a). We observed considerable potential for off-target activity in both LbCas12a and MAD7, only a single potential off-target for AsCas12a, and no off-targeting for the chimeric M44 (color screening assay) (Fig. 5a, Supplementary Fig. 14a–c). While these data were compelling, we further expanded these studies with a comprehensive and sensitive genome-wide off-target assay (CIRCLE-seq)[56,57]. Using *E. coli* MG1655 genome as targeting DNA, we tested 2 gRNAs (galK1 and lacZ2) using AsCas12a, LbCas12a, MAD7, and M44 (Fig. 5b, c). The target sequence is shown at the top of the figure (Fig. 5b, c) and off-target sequences are shown below. The read counts for each sequence are shown to the right and represent a measure of cleavage efficiency at a given site. For each gRNA, the target sequence is shown at the top of the figure (Fig. 5b, c) and off-target sequences are shown below (Fig. 5b, c). The read counts for each sequence are shown to the right and represent a measure of cleavage efficiency at a given site. M44 showed the same or less off-target activity when compared with AsCas12a and LbCas12a, and less off-target activity than MAD7. For the first guide, M44, AsCas12a, and LbCas12a all had 0% off-target activity, while MAD7 had 0.8% off-target activity (Fig. 5b). For the second guide, M44, AsCas12a, and LbCas12a, had 0, 7.8, and 23.7%, off-target activity, respectively, while MAD7 had more off-target activity than on-target (Fig. 5c). These results generally confirmed the results obtained using our designer off-target assay (Fig. 5b, c), MAD7 and LbCas12a had substantially higher off-targeting relative to AsCas12a and the chimeric M44.

**Chimeric nucleases enable genome editing in eukaryotic cell**. Targeted genome editing with the hope of treating and curing diseases has always been a major goal in the field of biology. Thus, we tested the chimeric Cas12a-like nucleases for genome editing in mammalian cells. We constructed a plasmid expressing the M44 chimeric nuclease (with T7 promoter), a single crRNA (with

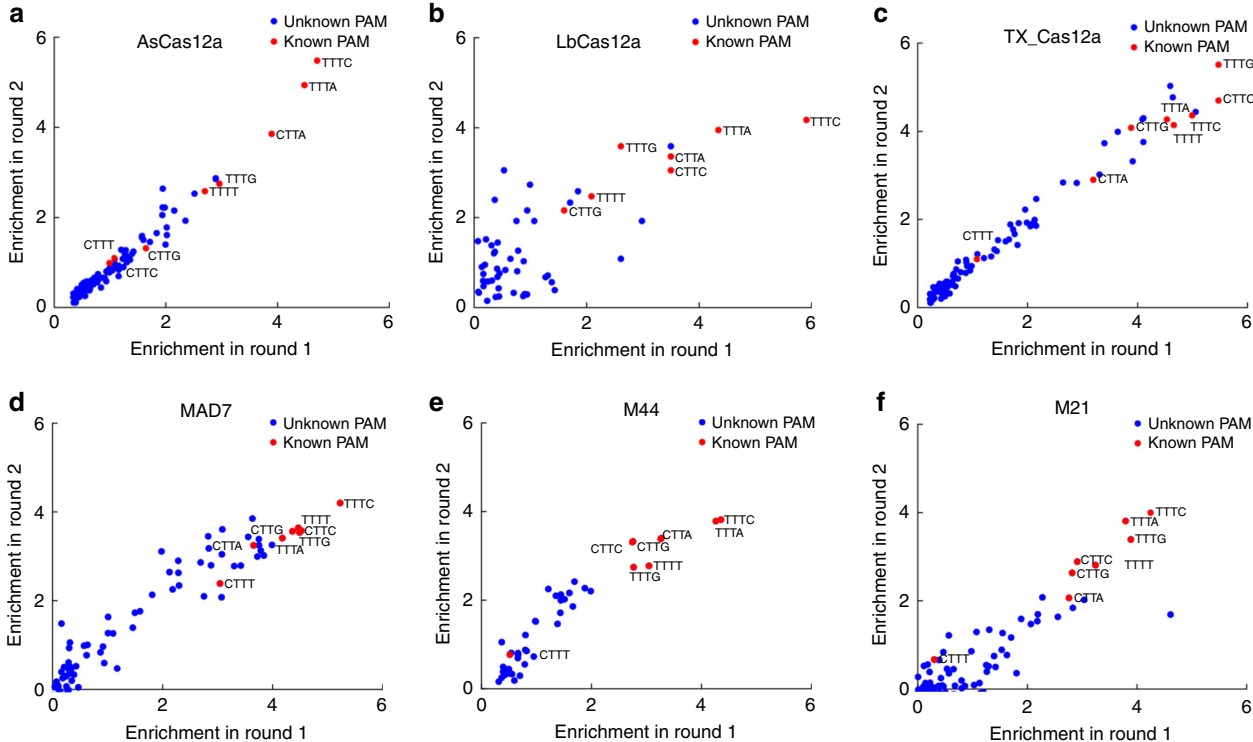

**Fig. 4** The specificity detection of chimeric Cas12a-type variants. **a–f** The enrichment score for two rounds of PAM scans. The enrichment score is the frequency change ($\log_2$) of each PAM using different gRNA plasmids (on-targeting and non-targeting gRNAs) (online method). **a** AsCas12a, **b** LbCas12a, **c** TX_Cas12a, **d** MAD7, **e** M44, and **f** M21.

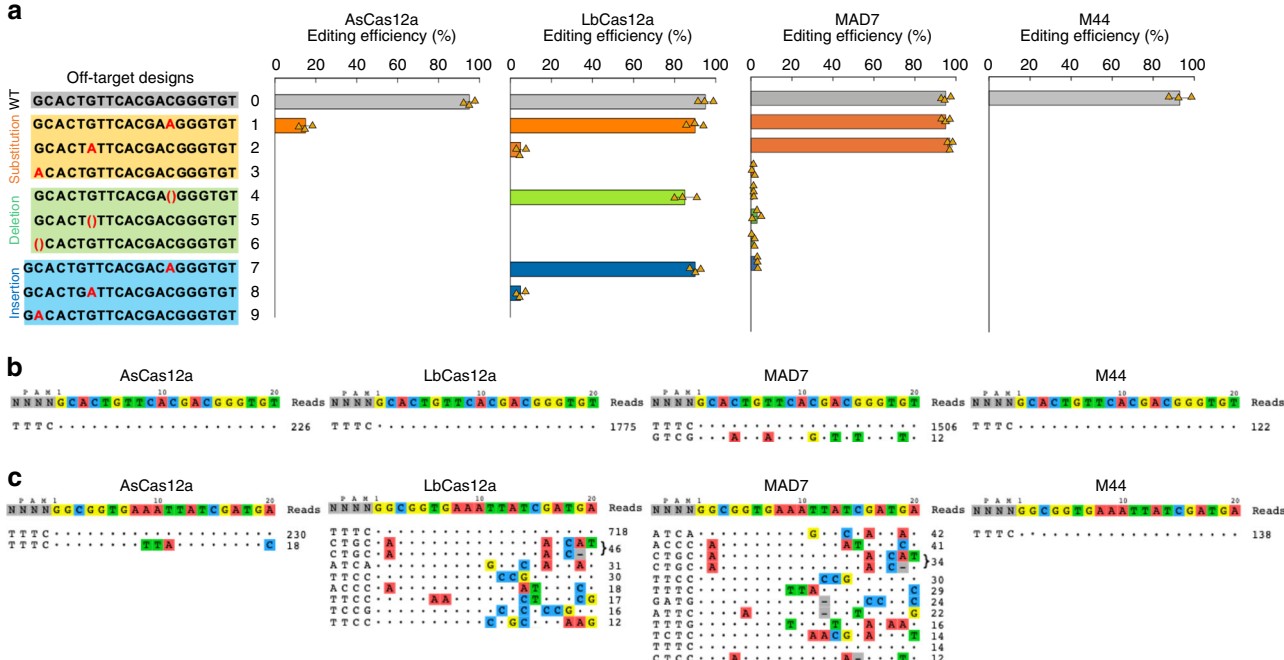

**Fig. 5** The off-target assay for chimeric Cas12a-type variants. **a** The individual off-target assay. We designed nine different off-target spacers, of which three were substitutions, three were deletions, and three were insertions. **b**, **c** Genome-wide off-target analysis was done using the CIRCLE-seq method[56,57]. **b** gRNA targeting the galK1 site and **c** gRNA targeting the lacZ2 site. Positions with mismatches to the target sequences, i.e., off-target sites, are highlighted in color. CIRCLE-seq read counts are shown to the right of the on- and off-target sequences and represent a measure of cleavage efficiency at a given site. The on/off-target reads shown in the figure were higher than 10. Source Data are available in the Source Data File.

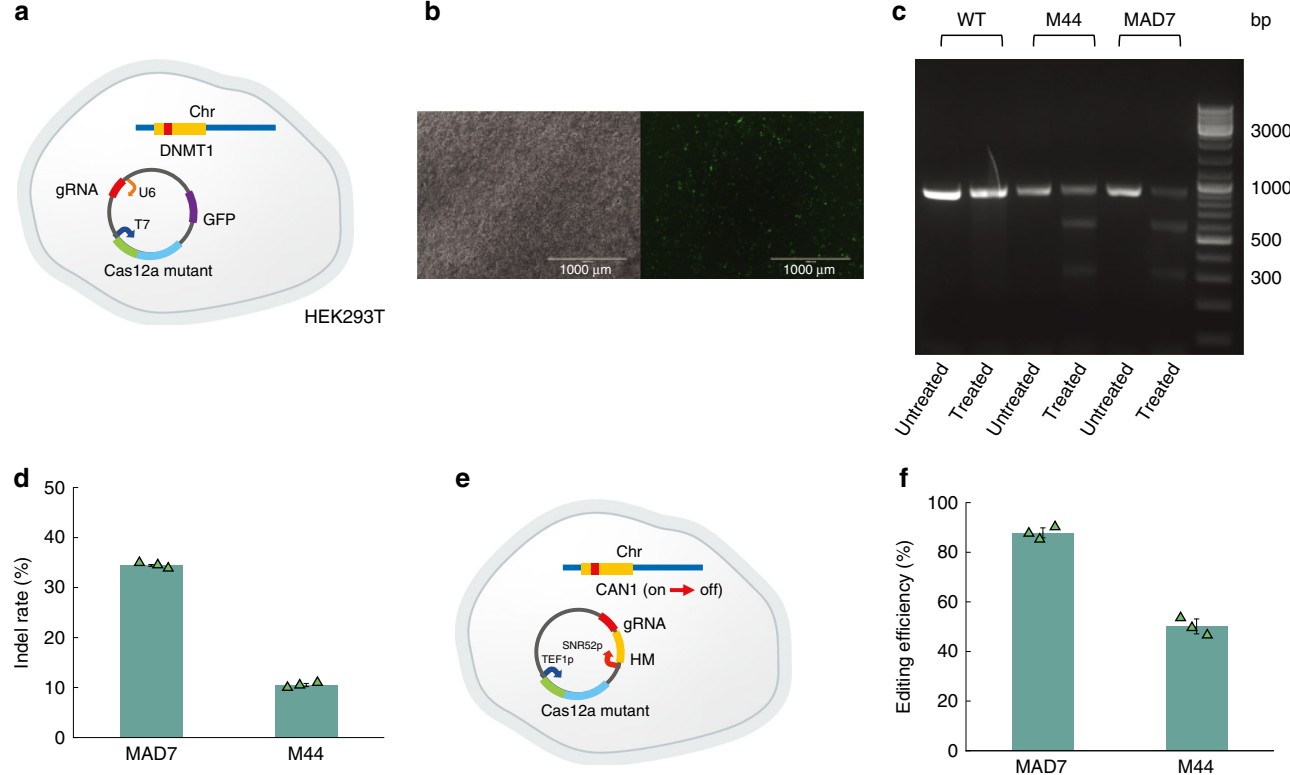

**Fig. 6** Chimeric nucleases enable genome editing in eukaryotic cells. **a** Genome editing in mammalian cells (HEK293T) using chimeric Cas12a-type variants. We constructed a plasmid expressing the M44 (or MAD7) nuclease (with T7 promoter), a single crRNA (with U6 promoter), and GFP. **b** Pictures of HEK293T after transfection. The H3K293T cells were transfected with the plasmid containing the M44 nuclease and GFP. Micrographs were taken under cool white light (left) or fluorescent light (right). **c** The T7E1 assay[52,58,59] was performed on cells that were expressing GFP and isolated by fluorescence-activated cell sorting. Untreated means the PCR products without T7 endonuclease treatment. Treated means the PCR products with T7 endonuclease treatment. **d** The indel rate of MAD7 and M44. The calculation was made using the formula shown in the methods section. **e** Genome editing in yeast (*S. cerevisiae* BY4741) using chimeric Cas12a-type variants. We constructed a plasmid containing the M44 (or MAD7) nuclease (with TEF1p promoter), a single crRNA (SNR52p promoter) targeting the *CAN1* gene and a homology arm (HM) containing a *CAN1*-inactivating mutation as a template for recombineering. Only colonies with an inactivated *CAN1* gene can grow on a +can plate. **f** The editing efficiency of MAD7 and M44. The editing efficiency was calculated by determining the ratio of colonies on plates +/−can. Editing was also confirmed by sequencing 20 colonies from +can plates. Source Data are available in the Source Data File.

U6 promoter), and GFP as a transfection control (Fig. 6a). Transfected cells were isolated by FACs (Fig. 6b), cell lysate harvested 72 h post-transfection, and indel detection was performed using T7E1 assay[52,58,59] (Fig. 6b, c). The results demonstrated that this chimeric nuclease is fully functional in mammalian gene editing experiments, although it has a lower editing efficiency than MAD7 (Fig. 6d) and AsCas12a[60].

In addition to mammalian cells, we further tested the chimeric Cas12a-type variants for genome editing in *S. cerevisiae* (Fig. 6e) *S. cerevisiae* has long been the most tractable organism for eukaryotic cell biology, owing to its genetic malleability, greatly facilitated by a preference for homologous recombination (HR) over non-homologous end joining (NHEJ) for double-stranded break (DSB) repair. To examine chimeric Cas12a-type nuclease activity in *S. cerevisiae*, we designed gRNAs to target the endogenous genomic negative selectable marker CAN1, which its null mutation can be selected with media containing canavanine (a toxic arginine analog)[61]. We found M44 edited at efficiencies >40%, which is lower than the editing efficiency of ~86% seen with MAD7 (Fig. 6f). Collectively, these results demonstrated that the chimeragenesis strategy reported can be used to rapidly develop synthetic nucleases with altered characteristics that are functional across multiple species.

## Discussion

Structure-guided chimeragenesis is an effective way to generate synthetic protein families with broad sequence diversity while maintaining a relatively high percentage of folded and functional proteins[34,35]. Furthermore, the proportion of folded variants can be increased through simple solutions such as utilizing stabilized parental sequences. Large datasets are generated by characterizing these libraries, and, unlike natural protein families, these sets include both functional and nonfunctional sequences that can be queried for specific properties in high-throughput formats[62]. There are abundant Cas12a-like family proteins in the database. However, not all the characterized Cas12a-like proteins are efficient in the model systems, such as *E. coli*, yeast, and mammalian cells[15], emphasizing the need to improve existing nuclease classifications. While many Cas12a-like sequences are easily identified, predicting which ones are functional and what are the preferred PAM designs and gRNA designs remains intractable. Here, we designed a Cas12a-type nuclease library with 560 mutants using domain recombination points based on the homology substructure of nine different Cas12as, and then to account for the lack of predictability we developed a selection/screening system to identify functional Cas12a-type chimeras. Interestingly, we observed that ~30% sequences of the positive

variants do not align to the WT Cas12a proteins. In addition, the PAM specificity and off-targeting characteristics were different among the characterized chimera mutants. Our approach therefore not only emphasizes the unique status of such nucleases but also the potential of this strategy for generating nucleases fit for a particular application.

Genome-editing efficiency and off-target effects are major challenges for the broad application of CRISPR systems[63,64]. Recently, several groups have reported that tailored substitutions abolishing nonspecific contacts between SpCas9 and the DNA substrate generate more precise RNA-guided endonucleases (eSpCas9(1.1)[25], evoCas9[55], and SpCas9–HF1[26]) with increased dependency on sgRNA:target DNA pairing. Even though these variants offer improved specificity, for some sites, off-target cleavage remains a problem[25,26,54,55]. Here, we have discovered several chimeric Cas12a-type variants with reduced off-target activity yet well-preserved on-target activity as demonstrated in vitro and in vivo in E. coli. We further showed that the chimeric Cas12a-like nucleases facilitate genome editing in E. coli, yeast, and mammalian cells, although editing efficiency in both yeast and mammalian cells is lower with the M44 chimera than with other nucleases[60] (Fig. 6d, f). While the M44 chimeric nuclease could benefit from further optimization to increase on-target editing in eukaryotic cells, there are also instances (e.g., allogenic cell therapies) in which having lower off-target activity may be a higher priority than having high on-target activity. We believe these chimeric nucleases could be beneficial in downstream applications ranging from human health to industrial biotechnology.

Engineering Cas12a-like nucleases based on structural information expands the current genome-editing toolbox[29]. For example, future efforts to further alter PAM preference or on/off-target specificity could entail the evaluation of a much larger collection of REC1 domains or by saturation/random mutagenesis of specific regions of chimeric Cas12a-like nucleases, among other well established directed evolution approaches. Furthermore, our strategy can be adapted to engineer additional homologous and non-homologous RNA-guided endonucleases. Finally, our screening platform could be applied for the development of chimeric Cas12a-type variants tailored to a range of specific functional objectives, such as optimization of editing at a specific loci or in a targeted cell line, among others. The vast collection of already identified nucleases in combination with a rapid approach for generating large combinations thereof opens up the potential for generation of specialized synthetic nucleases tailor made for a range of applications.

## Methods

**Chimeric Cas12a-type nuclease library construction.** Using Cas12a-type nuclease sequences available from the NCBI database, we performed alignments (Supplementary Fig. 2) to determine homologous domains to design chimera library sequences (Table S1). We used a ~40 bp homology arm with MAD7 and its plasmid as shown in Supplementary Fig. 1. All the library sequences (Table S1) were ordered as separate gBlocks from IDT (Integrated DNA Technologies, Coralville, IA). The library plasmid construction for 1 and 2 crossovers were used Gibson assembly method. The assembly used NEBuilder® HiFi DNA Assembly Master Mix (New England Biolabs, Ipswich, MA, USA), and the assembly products were desalted using dialysis by spotting the reaction on a filter with 0.025 μm pores floating in ddH₂O. Following desalting, the assembly products were electroporated into E. cloni 10G ELITE Electrocompetent Cells (Lucigen Corporation, Middleton, WI, USA). Libraries were spot plated onto LB with 34 μg/mL chloramphenicol to estimate transformation efficiency. The library plasmids were purified using QIAprep Spin Miniprep Kit (Qiagen, Valencia, CA, USA). All PCR steps were performed with the high-fidelity Phusion enzyme (New England Biolabs, Ipswich, MA, USA) to ensure production of a high-quality library.

**Nuclease-mediated cell killing assay.** We constructed a two plasmid system for genome editing, which expresses a Cas12a-like protein and a single crRNA (with J23119 promoter) targeting the galK or lacZ gene. For each experiment, we transformed equal amounts of non-targeting and on-targeting (e.g., galK1) gRNA plasmids. The cutting efficiency was calculated as following:

$$\text{Cutting efficiency} = \left(1 - \frac{a}{b}\right) \times 100\%$$

The same amount of culture was plated in two LB agar plates with chloramphenicol and carbenicillin. a denotes the number of colonies that can grow on the plate with on-targeting gRNA plasmid, and b is the number of colonies that can grow on the plate with non-targeting gRNA plasmid.

**Generation of heterologous plasmids.** To generate the Cas12a locus for heterologous expression, the Cas12a-type DNA sequences after codon optimization was PCR amplified and cloned into pSC101, pX2, pMINR[65], and pY094 using Gibson cloning kit (New England Biolabs). Sequences of all the chimera and gRNA design can be found in Supplementary Data 1.

**The isolation of functional Cas12a-type mutants.** The host strain carried the plasmid expressing lambda red proteins and chimeric Cas12a-like proteins library. The strain were cultured in 30 °C and supplemented with 0.2% arabinose for inducing lambda red proteins. When OD₆₀₀ reached 0.5–0.6, the cells were induced for 15 min at 42 °C to induce chimeric Cas12a-like proteins. After chilling on ice for 15–30 min, the cells were washed twice with 20% of the initial culture volume of ddH₂O. Then, the gRNA plasmid was mixed with the cells, followed by chilling on ice for 5 min. Following electroporation, the cells were recovered in SOB medium for 3 h. Then, 1 μL of cells was plated in the M9 agar media supplemented with 2-deoxy-galactose (DOG).

The isolation of functional Cas12a-type mutants directly in vivo potentially enabled the identification of Cas12a-type variants with higher editing efficiency. The galK gene product, galactokinase, catalyzes the first step in the galactose degradation pathway, phosphorylating galactose to galactose-1-phosphate. Galactokinase also efficiently catalyzes the phosphorylation of a galactose analog, 2-deoxy-galactose (DOG). The product of this reaction cannot be further metabolized, leading to a toxic build-up of 2-deoxy-galactose-1-phosphate[38]. Thus, strains with galK inactivation can grow in the media supplementary with 2-DOG and background following negative selection is reduced and no colony screening is necessary.

The selected Cas12a-type mutants were verified using the above competent cell preparation and transformation method. After 3 h recovery, 1 μL of cells was plated in the MacConkey agar. The color screening method based on the galK inactivation to evaluate the editing efficiency of CRISPR–Cas9 was same as the previous studies[7,37].

**Cas12a PAM screen.** PAM plasmid libraries were constructed using synthesized oligonucleotides (IDT) containing the designed NNNN PAM library. The dsDNA product was assembled into a linearized plasmid (containing kanR gene) using Gibson cloning (New England Biolabs). The PAM library was transformed into MG1655 with the plasmid expressing chimeric Cas12a-like proteins using the electroporation method. We then transformed two equivalent gRNA plasmids individually into the E. coli MG1655. One gRNA design is targeted on the library sites, and another gRNA plasmid is non-targeting control. We collected the cells grown on kanamycin media using different gRNA plasmids, and amplified the region of the PAM library from the reported plasmid for the high-throughput sequencing. The enrichment score of PAM and accompanying sequence logo for one of two library replicates are shown in PAM screening revealed the PAM specificity were different between different chimeric Cas12a-like proteins. The prepared cDNA libraries were sequenced on a MiSeq with a single-end 300 cycle kit (Illumina). Indels were mapped using a Python implementation of the Geneious 6.0.3 Read Mapper.

$$E_i = \frac{\log(Y_i)}{\log(X_i)}$$

$E_i$ denotes the enrichment score. $X_i$ is the frequency of PAM i using on-targeting gRNA plasmid in the deep sequencing measurements. $Y_i$ is the frequency of PAM i using non-targeting gRNA plasmid in the deep sequencing measurements.

**Yeast transformation.** High-efficiency yeast transformation was conducted using the LiAc/SS carrier DNA/PEG method[66]. Briefly, 50 ml of YPD were inoculated with 500 μL of the YPD overnight culture in a 250 ml flask. The cells (OD = 0.6) were harvested by centrifuge at 4000 rpm (3130 g) for 5 min. Pellets were resuspended in 25 ml of sterile water by vortexing briefly and then resuspended in 1 ml of 100 mM LiAc. The cell suspension was transferred to a 1.5 ml tube, and centrifuged at 3000 g for 2 min, after which the supernatant was discarded. 400 μl 100 mM LiAc was added and the cells resuspended. 50 μL of this was aliquoted into 1.5 ml tubes (1 for each transformation). Cells were pelleted (3000 g for 2 min) and the supernatant removed. The following was added to each 1.5 ml tube of cells; 300 μL T mix (240 μL 50% PEG 3350, 35 μL 1.0 M lithium acetate, 25 μL 2 mg/ml sssDNA), 50 μL sterile H₂O, and 20 μL of DNA (1 μg). After vortexing to resuspend cells, the tubes were incubated for 30 minutes at 30 °C, and next incubated in a water bath at 42 °C for 20-25 (up to 40) min. Tubes were microfuged at 3000 g for

15 s, and the transformation mix removed with a micropipette. 200 μL of sterile water was added to each tube and cells resuspended by pipetting it up and down. Finally, 150 μl of sterile water was plated and 20 μl cell suspension added in one selection plate, to be incubated at 30 °C.

**PEI transfection**. HEK293T (ATCC® CRL-3216™) were cultured in 6-well dish with 60% confluency. After cells attached on the surface of the dish, for each well, two 1.5 mL centrifuge tubes were loaded with 250 μL serum-free and phenol red-free DMEM. One of the tubes was loaded with 3 μL of polyehtyleimine (PEI, concentration: 1 mg/mL), and the other one tube was loaded with 1 μg of plasmid. After addition, tubes were mixed and placed for 4 min. After placing, tubes loaded with PEI were mixed to tubes with specific plasmid drop-wisely. Tubes were placed for 20 min after mixing and mixtures were added into wells drop-wisely.

**Fluorescence-activated cell sorting (FACS)**. HEK293T was incubated with 1 mL (0.5%) trypsin at 37 °C for 5 min followed by pelleting and resuspension in DMEM with 5% fetal bovine serum (FBS). Resuspended cells were filtered with CellTrics® 50 μm filter to discard debris. Cell sorting was performed using BD FACSAria™ Fusion equipped with OBIS 488 nm laser (SN: 177745) at 98.3 mW of power. Forward scatter area (FSC-A), side scatter area (SSC-A) and side scatter width (SSC-W) were collected through a filter. The GFP signal was collected in the 488 nm channel through a 530/30-A band pass filter. The first gate was drawn in the SSC-A/FSC-A plot to include cells with universal size, and the second gate was drawn in the SSC-A/SSC-W plot to include single cells. The third gate was drawn in the FSC-A/488 B 530/30-A channel to sort cells with GFP signal.

**T7E1 assay**. Genomic DNA was extracted using the QuickExtract DNA Extraction Solution (Epicenter) following the manufacturer's protocol. The genomic region flanking the CRISPR target site for each gene was PCR amplified, and products were purified using QiaQuick Spin Column (QIAGEN) following the manufacturer's protocol. 200–500 ng total of the purified PCR products were mixed with 1 μl 10 × Taq DNA Polymerase PCR buffer (Enzymatics) and ultrapure water to a final volume of 10 μl and were subjected to a re-annealing process to enable heteroduplex formation: 95 °C for 10 min, 95–85 °C ramping at −2 °C/s, 85–25 °C at −0.25 °C/s, and 25 °C hold for 1 min. After re-annealing, products were treated with T7 Endonuclease I (NEB) following the manufacturer's recommended protocol and analyzed on 4–20% Novex TBE polyacrylamide gels (Life Technologies). Gels were stained with SYBR Gold DNA stain (Life Technologies) for 10 min and imaged with a Gel Doc gel imaging system (Bio-Rad). Quantification was based on relative band intensities. Indel percentage was determined by the formula, $100 \times (1 - \mathrm{sqrt}(1 - (b + c)/(a + b + c)))$, where $a$ is the integrated intensity of the undigested PCR product, and $b$ and $c$ are the integrated intensities of each cleavage product.

**Reporting summary**. Further information on research design is available in the Nature Research Reporting Summary linked to this article.

## Data availability

Data supporting the findings of this manuscript are available from the corresponding authors upon reasonable request. The source data underlying Figs. 1, 2, 3, 5, 6 and Supplementary Figs. 7, 8, 10, 11, 13 are provided as a Source Data file.

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

## Acknowledgements

The project was sponsored by the US Department of Energy (Grant DE-SC0018368).

## Author contributions

R.L. and R.T.G. developed the concept. R.L., L.L., C.E. and A.D.G. all aided in the design of experiments. The experiments of *E. coli* were done by R.L., L.L., H.C. and F.E. The experiments of yeast were done by L.L. and O.E. The experiments of mammalian cells were done by R.L., F.E. and Z.L.

## Competing interests

The authors declare the following competing interests: R.T.G. and A.G. have financial interests in Inscripta, Inc.
