## [Peer Review File · Nature Communications]

Reviewers' Comments:

Reviewer #1:

Remarks to the Author:

In this work, Liu et al. describe a strategy to generate many chimeric Cas12a nucleases using a domain swapping approach. From a possible starting library of 560 mutants from 9 parent sequences, they identify 24 unique chimeric Cas12a sequences capable of editing DNA in a galK-based assay. Moreover, they further characterized several variants and found that they demonstrate altered PAM specificity and the ability to function in yeast and HEK293T cells. The overall results presented in this work should be of interest to a broad range of readers and provide a method by which chimeric variants of other Cas nucleases could be developed. However, this manuscript as written is often unclear; in particular, the figure captions throughout the manuscript are insufficient to guide the reader through the results of the work. Substantial text revisions are therefore needed to improve the clarity of the manuscript.

Major comments:

- Throughout the manuscript, figure captions are insufficient to inform the reader of important details, results, and conclusions of the data presented in the figure.
- A more thorough discussion of the design process and its limitations (outlined in figure 1) would be useful for the reader. In particular, it is not clear from the figure or text why those particular crossover points in figure 1c were chosen. Why were other locations – for example, the WED-II/PI junction – not used as a crossover point? Additionally, was there a reason that more than 2 crossover points were not investigated?
- No chimera was developed with improved cutting ability over the MAD7 starting variant (Figure 2b), which slightly reduces the impact of this work.

Minor comments:

- Line 50: use of the word 'possible' is a little unclear. It may be worth briefly mentioning how the off-targeting profiles of these nucleases differ from Cas9
- Line 56: see previous comment
- Lines 64, 67, and 71: "et al" should read "et al."
- Line 106: A discussion of the galK inactivation assay (both editing efficiency and the CFU/ug measure) is merited here.
- Fig 1b is mentioned before figure 1a. In general, the order of elements in the main text should match their order in the figures.
- Line 124: identification of crossover points needs more detail
- Line 134: "chimera" should read "chimeric"
- Line 134-136: the conclusions drawn from the two ideas in this sentence seem to be mixed up; shouldn't a small number of sequences not suggest that all domains can be recombined to produce a spectrum of activities?
- Line 141-143: This statement discussing the crossover at position 6 should be expanded on.
- Line 145: figure 2a,2b should be cited (not 2b,2c)
- Line 153: should read "positioned in the galK and lacZ genes..."
- Line 154-155: this nuclease-mediated cell killing assay should be made more clear to the reader, as it is not made clear in the caption of figure 3 either.
- Line 158/Figure 3b: possible reasons for the observed differences in cutting efficiency at different sites within the same gene should be discussed.
- Line 161: Possible explanations for differences in transformation efficiency should be discussed. For example, based on figure 2b it appears that more functional nucleases have lower transformation efficiencies; does increased editing efficiency result in decreased fitness?
- Line 165: "safe site" should read "safe sites"
- SI Figure 7 caption: "position" should read "positions"
- SI Figure 7: the position of the origin of replication should be more clearly shown. It may be useful to the reader to note the distance from the origin in parts b and c
- Line 189: "Firstly" should read "first"

- Line 244: the enrichment score should be explained
- Line 245: "1" should be spelled "one"
- Line 270: Cas12a should be capitalized
- Line 274: the T7E1 assay should be referenced or briefly discussed.
- Line 360: "consisting of" should read "containing the"
- Line 362: "were" should read "was", "expressed" should read "expressing"
- Line 363: "of" should be omitted
- Line 387: "was" should read "were"
- Line 559: should read "targeting the galK gene"
- Line 560: "Supplementary" should read "supplemented"
- Line 562: This statement is unclear to the reader.
- Line 564: the transformation efficiency test should be described; it is unclear what is being tested.
- The supplementary table as presented is cumbersome; an excel file may be more useful.

Reviewer #2:

Remarks to the Author:

In this study, Liu et al. construct a library of Cas12a chimeras (560 chimeras) to screen for functional synthetic variants. They have successfully identified functional chimeras, albeit with lower on-target activity than an established WT Cas12a enzyme. The authors profile the difference in PAM preferences of 3 of the functional chimeras and perform basic off-target analyses for 2 of the enzymes. More data is needed to understand whether any of these chimeras are preferable to established Cas12a enzymes (e.g. MAD7, Lb, As). Although the screening technique is interesting, the work requires more in-depth validation of the new synthetic enzymes.

Specific comments:

* The authors based the library design on nine Cas12a variants and stated that they are distally related with a low level of sequence homology. Since the gene editing community are more familiar with AsCas12a and LbCas12a, the authors should include these variants in their phylogenetic analyses. Also, I would be curious to see what is the sequence homology quantitatively among all nine variants (plus As and LbCas12a).

* How are the six crossover points identified/chosen? Figure 1C shows ten junctions of eleven functional domains, but only six are labeled as crossover points. Are the junctions/crossover-points conserved among nine variants? It might be helpful to show the sequence alignment of the starting 9 enzymes (perhaps as a supplementary figure) to better explain how the six crossover points are identified.

* No description is given of how the chimera library is created. Was it synthesized? Was it cloned through a recombineering technique? What was the starting material? Given that the library is a major focus (if not, THE major focus) of the paper, it is surprising that the authors omit this. A detailed methods section on library design is a necessity (and even more so given their affiliation/competing financial interest with a commercial Cas12a company, Inscripta).

* It is unclear to me why it is important to create Cas12a chimeras with a spectrum of low on-target activities and undefined PAMs. The authors have suggested that variants can be used for "optimization of editing at a specific loci or in a targeted cell line", however, no direct evidence is shown in the paper (the only one case where synthetic chimera seems to perform better, is in supplementary fig10d, with GTTA PAM, M44 has > 90% cutting efficiency whereas MAD7 has ~60%). If the goal is to engineer a better Cas12a variant with either higher on-target activity, or lower off-target activity, or more flexible PAMs, the study might benefit from comparing with the well-characterized Cas12a variants (AsCas12a and LbCas12a) and the widely-used SpCas9.

* The authors have developed a high-throughput in vivo screen to profile the PAM preference of the selected three variants. Given that it is a novel method; it would be very helpful to explain clearly the workflow and how the enrichment score is calculated, and provide some quality control data of the screen. Are Fig 4a-d plotting enrichment score (the text cited Fig 4a-d and suggested it is enrichment score, however, the scale appears to be a read count)?

* Fig 4e-h: It is great to see the majority of enrichment scores in round 1 are consistent with round 2, indicating high replicability. And it is interesting to see that M44 has less unknown PAMs highly enriched, as compared to MAD7. Since M44 is a chimera of TX_Cas12a REC1 domain and the rest of MAD7, I wonder if M44 is simply adopting TX_Cas12a's PAM preference. To aid in interpretation of this data, the authors should include the same assay for TX_Cas12a.

* Since novel PAM targeting is the main take-away for the genome engineering community, the authors should also perform the PAM assay on AsCas12a and LbCas12a. These are the most commonly used Cas12a and thus comparison with these enzymes is an absolute requirement for claiming superior properties of the new chimeric enzymes.

* For off-target effects, the study created nine possible off-target binding sites from one WT binding site, and suggested M44 mutant has lower off-target effect compare to WT MAD7. From Kim et al, NBT, 2017, we learned that AsCas12a/LbCas12a guides have a 'seed region' where altering any bases of either position 1-6 will abolish the cleaving activity almost entirely. It is interesting to see MAD7 retains the full cleaving potential with an altered base on position 6. This suggests that MAD7 might be a less specific variant than AsCas12a. However, with only one synthetic target site, it is not sufficient to conclude whether MAD7 is a less specific variant or not. Assuming MAD7 indeed has this off-target concern (and it is likely given that Fig 4e suggests that MAD7 can recognize many unknown PAMs) and we know from Kim et al. 2017 and Kleinstiver et al. 2016 that AsCas12a and LbCas12a have extremely low genome-wide off-target rates, it is questionable that the new M44 mutant will perform better than the existing tools in terms of off-target rate.

* Off-target analyses should be done genome-wide (e.g. GUIDE-seq, BLISS/BLESS, etc.). The authors use a very small number of mismatched guides and targets (< 10). In 2019, this is not acceptable for accurate quantification of off-targets given the abundance of genome-wide techniques.

* Overall, I feel that it is possible that the authors have discovered a superior Cas12a enzyme (along some dimension) but unfortunately the PAM and off-target analyses do not convincingly demonstrate superiority over established Cas12a (e.g. AsCas12a, LbCas12a) or over the source Cas12a for chimeras M44 and M21 (TxCas12a). These comparisons would greatly improve the manuscript.

Minor comments:

* For Fig1b figure annotations, to increase readability, it would be helpful to either specify which color represents editing efficiency/transformation efficiency in the figure legend or use more obviously distinguishable colors.

* Fig1h, the titles of the bar graphs need to be updated (according to text, the left graph is MAD7, right is M44)

* There is no financial interest disclosure. Given that one author is affiliated with a company whose primary product is Cas12a ("MAD7"), any potential or perceived conflicts should be disclosed in detail.

Dear editors and reviewers,

Thank you very much for your comments concerning our manuscript entitled “**Synthetic chimeric nucleases function for efficient genome editing**”. The comments were helpful for revising and improving our paper. We have studied the comments carefully and we made appropriate revisions. The main corrections are marked in red in this revised manuscript and our detailed responses to the comments are given below.

Reviewers' comments:

Reviewer #1 (Remarks to the Author):

In this work, Liu et al. describe a strategy to generate many chimeric Cas12a nucleases using a domain swapping approach. From a possible starting library of 560 mutants from 9 parent sequences, they identify 24 unique chimeric Cas12a sequences capable of editing DNA in a galK-based assay. Moreover, they further characterized several variants and found that they demonstrate altered PAM specificity and the ability to function in yeast and HEK293T cells. The overall results presented in this work should be of interest to a broad range of readers and provide a method by which chimeric variants of other Cas nucleases could be developed. However, this manuscript as written is often unclear; in particular, the figure captions throughout the manuscript are insufficient to guide the reader through the results of the work. Substantial text revisions are therefore needed to improve the clarity of the manuscript.

Major comments:

- Throughout the manuscript, figure captions are insufficient to inform the reader of important details, results, and conclusions of the data presented in the figure.

Reply: We have modified the figure captions to include more detail.

- A more thorough discussion of the design process and its limitations (outlined in figure 1) would be useful for the reader. In particular, it is not clear from the figure or text why those particular crossover points in figure 1c were chosen. Why were other locations – for example, the WED-II/PI junction – not used as a crossover point? Additionally, was there a reason that more than 2 crossover points were not investigated?

Reply: We have modified the related text (page 5 line 125-138), and added a new figure (Supplementary Fig. 2) to address this. During library construction, mutants with more than 2 crossover points did not show any activity. Thus, they were eliminated from further investigation.

- No chimera was developed with improved cutting ability over the MAD7 starting variant (Figure 2b), which slightly reduces the impact of this work.

Reply: In the figure 3b, the cutting efficiency of the variants are higher than MAD7 in the lacZ2 and lacZ3 experiments. In the figure 3c, the editing efficiency of the variants are higher than MAD7 in the galk2 and lacZ2 experiments. We have modified the related text on page 6 lines 171-178. In addition, the variants have reduced off-targeting relative to MAD7, AsCas12a, and LbCas12a.

Minor comments:

- Line 50: use of the word 'possible' is a little unclear. It may be worth briefly mentioning how the off-targeting profiles of these nucleases differ from Cas9

Reply: We have described it clearly in the text (Page 3 line 70-72).

- Line 56: see previous comment

Reply: We have described it clearly in the text (Page 3 line 70-72).

- Lines 64, 67, and 71: "et al" should read "et al."

Reply: We have modified it.

- Line 106: A discussion of the galk inactivation assay (both editing efficiency and the CFU/ug measure) is merited here.

Reply: We have modified it (Page 4 line 121-123).

- Fig 1b is mentioned before figure 1a. In general, the order of elements in the main text should match their order in the figures.

Reply: The fig 1a is mentioned in the introduction section before result section (Page 2 line 55).

- Line 124: identification of crossover points needs more detail

Reply: We have modified it (page 5 line 125-138).

- Line 134: "chimera" should read "chimeric"

Reply: We have modified it.

- Line 134-136: the conclusions drawn from the two ideas in this sentence seem to be mixed up; shouldn't a small number of sequences not suggest that all domains can be recombined to produce a spectrum of activities?

Reply: We have modified it (page 5 line 145-147).

- Line 141-143: This statement discussing the crossover at position 6 should be expanded on.

Reply: We have modified it (page 6 line 155-157).

- Line 145: figure 2a,2b should be cited (not 2b,2c)

Reply: We have modified it.

- Line 153: should read “positioned in the galK and lacZ genes...”

Reply: We have modified it.

- Line 154-155: this nuclease-mediated cell killing assay should be made more clear to the reader, as it is not made clear in the caption of figure 3 either.

Reply: We have added the details for the assay in the method section (page 13 line 374-383)

- Line 158/Figure 3b: possible reasons for the observed differences in cutting efficiency at different sites within the same gene should be discussed.

Reply: We have added possible reasons (page 6 line 171-173)

- Line 161: Possible explanations for differences in transformation efficiency should be discussed. For example, based on figure 2b it appears that more functional nucleases have lower transformation efficiencies; does increased editing efficiency result in decreased fitness?

Reply: We have modified it (page 6 line 180-184).

- Line 165: “safe site” should read “safe sites”

Reply: We have modified it.

- SI Figure 7 caption: “position” should read “positions”

Reply: We have modified it.

- SI Figure 7: the position of the origin of replication should be more clearly shown. It may be useful to the reader to note the distance from the origin in parts b and c

Reply: We have modified it.

- Line 189: “Firstly” should read “first”

Reply: We have modified it.

- Line 244: the enrichment score should be explained

Reply: We have explained it in the modified figure legend.

- Line 245: “1” should be spelled “one”

Reply: We have modified it.

- Line 270: Cas12a should be capitalized

Reply: We have modified it.

- Line 274: the T7E1 assay should be referenced or briefly discussed.

Reply: We added several references for this assay.

- Line 360: “consisting of” should read “containing the”

Reply: We have modified it.

- Line 362: “were” should read “was”, “expressed” should read “expressing”

Reply: We have modified it.

- Line 363: “of” should be omitted

Reply: We have modified it.

- Line 387: “was” should read “were”

Reply: We have modified it.

- Line 559: should read “targeting the galK gene”

Reply: We have modified it.

- Line 560: “Supplementary” should read “supplemented”

Reply: We have modified it.

- Line 562: This statement is unclear to the reader.

Reply: We have modified it.

- Line 564: the transformation efficiency test should be described; it is unclear what is being tested.

Reply: We have modified it (page 22 line 648 to 649).

- The supplementary table as presented is cumbersome; an excel file may be more useful.

Reply: We have make a excel file for it.

Reviewer #2 (Remarks to the Author):

In this study, Liu et al. construct a library of Cas12a chimeras (560 chimeras) to screen for functional synthetic variants. They have successfully identified functional chimeras, albeit with lower on-target activity than an established WT Cas12a enzyme. The authors profile the difference in PAM preferences of 3 of the functional chimeras and perform basic off-target analyses for 2 of the enzymes. More data is needed to understand whether any of these chimeras are preferable to established Cas12a enzymes (e.g. MAD7, Lb, As). Although the screening technique is interesting, the work requires more in-depth validation of the new synthetic enzymes.

Specific comments:

* The authors based the library design on nine Cas12a variants and stated that they are distally related with a low level of sequence homology. Since the gene editing community are more familiar with AsCas12a and LbCas12a, the authors should include these variants in their phylogenetic analyses. Also, I would be curious to see what is the sequence homology quantitatively among all nine variants (plus As and LbCas12a).

Reply: We added AsCas12a and LbCas12a to our analysis. We have modified the related text (page 5 line 125-127). In addition, the phylogenetic analyses is shown in the Supplementary Fig. 3. We have also added Supplementary Table 2 for sequence homology analysis.

* How are the six crossover points identified/chosen? Figure 1C shows ten junctions of eleven functional domains, but only six are labeled as crossover points. Are the junctions/crossover-points conserved among nine variants? It might be helpful to show the sequence alignment of the starting 9 enzymes (perhaps as a supplementary figure) to better explain how the six crossover points are identified.

Reply: We have modified the related text (page 5 line 125-137), and added sequence alignment in Supplementary Fig. 2 (separated pdf file) to better explain the design.

* No description is given of how the chimera library is created. Was it synthesized? Was it cloned through a recombineering technique? What was the starting material? Given that the library is a major focus (if not, THE major focus) of the paper, it is surprising that the authors omit this. A detailed methods section on library design is a necessity (and even more so given their affiliation/competing financial interest with a commercial Cas12a company, Inscripta).

Reply: We have modified the method section (page 12 line 358 to page 13 line 372 and Supplementary Fig. 1) for the library construction. We have modified the financial interest in addition information (page 16 line 484).

* It is unclear to me why it is important to create Cas12a chimeras with a spectrum of low on-target activities and undefined PAMs. The authors have suggested that variants can be used for “optimization of editing at a specific loci or in a targeted cell line”, however, no direct evidence is shown in the paper (the only one case where synthetic chimera seems to perform better, is in supplementary fig10d, with GTTA PAM, M44 has > 90% cutting efficiency whereas MAD7 has ~60%). If the goal is to engineer a better Cas12a variant with either higher on-target activity, or lower off-target activity, or more flexible PAMs, the study might be benefit from comparing with the well-characterized Cas12a variants (AsCas12a and LbCas12a) and the widely-used SpCas9.

Reply: We agree and have added the related data for AsCas12a and

LbCas12a (page 9 line 264 to page 10 line 296). We found that M44 has lower off-target activity than both AsCas12a and LbCas12a. We additionally found that M44 and M21 have higher editing efficiency than MAD7 with some gRNAs in E. coli (Fig. 3c).

* The authors have developed a high-throughput in vivo screen to profile the PAM preference of the selected three variants. Given that it is a novel method; it would be very helpful to explain clearly the workflow and how the enrichment score is calculated, and provide some quality control data of the screen. Are Fig 4a-d plotting enrichment score (the text cited Fig 4a-d and suggested it is enrichment score, however, the scale appears to be a read count)?

Reply: We have modified the method section (page 14 line 413-430) and figure legend to explain the workflow.

* Fig 4e-h: It is great to see the majority of enrichment scores in round 1 are consistent with round 2, indicating high replicability. And it is interesting to see that M44 has less unknown PAMs highly enriched, as compared to MAD7. Since M44 is a chimera of TX_Cas12a REC1 domain and the rest of MAD7, I wonder if M44 is simply adopting TX_Cas12a's PAM preference. To aid in interpretation of this data, the authors should include the same assay for TX_Cas12a.

Reply: We performed such assays as suggested, and note that TX does not have the same low level of unknown PAMs highly enriched. We have added the same assay for TX_Cas12a (page 9 line 264-272).

* Since novel PAM targeting is the main take-away for the genome engineering community, the authors should also perform the PAM assay on AsCas12a and LbCas12a. These are the most commonly used Cas12a and thus comparison with these enzymes is an absolute requirement for claiming superior properties of the new chimeric enzymes.

Reply: Agreed, we have performed the PAM assay on AsCas12a and LbCas12a and modified it in the related text (page 9 line 264-272).

* For off-target effects, the study created nine possible off-target binding sites from one WT binding site, and suggested M44 mutant has lower off-target effect compare to WT MAD7. From Kim et al, NBT, 2017, we learned that AsCas12a/LbCas12a guides have a 'seed region' where altering any bases of either position 1-6 will abolish the cleaving activity almost entirely. It is interesting to see MAD7 retains the full cleaving potential with an altered base on position 6. This suggests that MAD7 might be a less specific variant than AsCas12a. However, with only one synthetic target site, it is not sufficient to conclude whether MAD7 is a less specific variant or not. Assuming MAD7 indeed has this off-target concern (and it is likely given that Fig 4e suggests that MAD7 can recognize many unknown PAMs) and we know from Kim et al.

2017 and Kleinstiver et al. 2016 that AsCas12a and LbCas12a have extremely low genome-wide off-target rates, it is questionable that the new M44 mutant will perform better than the existing tools in terms of off-target rate.

* Off-target analyses should be done genome-wide (e.g. GUIDE-seq, BLISS/BLESS, etc.). The authors use a very small number of mismatched guides and targets (< 10). In 2019, this is not acceptable for accurate quantification of off-targets given the abundance of genome-wide techniques.

Reply: We have performed genome-wide off-target analysis (CIRCLE-seq) and added this to the revised manuscript (page 9 line 273 to page 10 line 296). We found that M44 had lower genome-wide off-target activity than MAD7, AsCas12a, or LbCas12a

* Overall, I feel that it is possible that the authors have discovered a superior Cas12a enzyme (along some dimension) but unfortunately the PAM and off-target analyses do not convincingly demonstrate superiority over established Cas12a (e.g. AsCas12a, LbCas12a) or over the source Cas12a for chimeras M44 and M21 (TxCas12a). These comparisons would greatly improve the manuscript.

Reply: We have added comparisons to AsCas12 and LbCas12a for the PAM and off-target analysis, as discussed above. We believe our new data show superior characteristics for one chimera (decreased off-target activity for M44).

Minor comments:

* For Fig1b figure annotations, to increase readability, it would be helpful to either specify which color represents editing efficiency/transformation efficiency in the figure legend or use more obviously distinguishable colors.

Reply: We have modified the figure caption.

* Fig1h, the titles of the bar graphs need to be updated (according to text, the left graph is MAD7, right is M44)

Reply: We have modified it.

* There is no financial interest disclosure. Given that one author is affiliated with a company whose primary product is Cas12a ("MAD7"), any potential or perceived conflicts should be disclosed in detail.

Reply: We have modified it in additional information (page 16 line 484).

Reviewers' Comments:

Reviewer #1:

Remarks to the Author:

The authors added significant amounts of additional clarifying text and new experiments to address the reviewer comments, including details of library design and a new genome-wide assay to assess off-target effects. I think the paper is now suitable for publication.

Reviewer #2:

Remarks to the Author:

It is great to see many details being added and some of the points regarding to the experimental design being addressed. However, overall, I feel that the readability and clarity of the manuscript still needs improvement.

Regarding the superiority of the chimera M44, the data presented in the manuscript is insufficient to support the claim of 'decreased off-target activity for M44'. In addition, for the statement 'we have discovered several chimeric Cas12a-type variants with substantially eliminated off-target activity and well-preserved on-target activity' (page 12, line 340-341), 'in E.coli' should be added for clarity. The authors need to deliver the information rather directly that the genome-wide off-target comparison is done in E.coli only and with only two guides (Figure 5c). Among those two guides, one showed the same off-target profile (zero off-target rates) between known Cas12as and the tested variant. The other guide showed 7.3% off-target rates for AsCas12a and 0% for M44.

Also, the on-target efficiency was compared to Cas9 (which cannot recognize T-rich PAMs at all, Figure 3C, making this a rather unfair comparison). The more appropriate comparison would be to As/LbCas12as, which do recognize this PAM.

From Figure 6D, in mammalian system, the editing rates of M44 (~10%) is lower than wild-type MAD7 (~35%), and also appears to be lower than previously reported AsCpf1+engineered crRNA system (~50% across three different cell lines and multiple locus, Li et al., Nature Biomedical Engineering, 2017, PMID: 28840077). Although it is not a side-by-side comparison with the same targeting site, the reported editing rate is lower than the current state-of-the-art. Given that M44 on-target activity is 3-fold lower than wild-type MAD7, this seems to suggest that the chimeric strategy might not be able to yield better Cas variants. The specificity might be improved slightly here with the new variants, but this might just be because of the overall lower on-target efficiency.

Overall, I am unconvinced the M44 offers any real improvement. This negative result may merit publication but the presentation should be changed to emphasize this. Otherwise, the gene editing community may invest significant time and effort in an enzyme that is inferior to current Cas12 enzymes.

Dear editors and reviewers,

Thank you very much for your comments concerning our manuscript entitled “**Synthetic chimeric nucleases function for efficient genome editing**”. The comments were helpful for revising and improving our paper. We have studied the comments carefully and we made appropriate revisions. The main corrections are marked in red in this revised manuscript and our detailed responses to the comments are given below.

Reviewers' comments:

Reviewer #1 (Remarks to the Author):

The authors added significant amounts of additional clarifying text and new experiments to address the reviewer comments, including details of library design and a new genome-wide assay to assess off-target effects. I think the paper is now suitable for publication.

Response: We thank the reviewer for her/his valuable comments which helped to considerably improve the quality of the manuscript.

Reviewer #2 (Remarks to the Author):

It is great to see many details being added and some of the points regarding to the experimental design being addressed. However, overall, I feel that the readability and clarity of the manuscript still needs improvement.

Regarding the superiority of the chimera M44, the data presented in the manuscript is insufficient to support the claim of ‘decreased off-target activity for M44’. In addition, for the statement ‘we have discovered several chimeric Cas12a-type variants with substantially eliminated off-target activity and well-preserved on-target activity’ (page 12, line 340-341), ‘in E.coli’ should be added for clarity.

Response: We appreciate the comment and have edited the text to now read - -“We have discovered several chimeric Cas12a-type variants with reduced off-target activity yet well-preserved on-target activity as demonstrated in-vitro and in-vivo in E. coli”.

The authors need to deliver the information rather directly that the genome-wide off-target comparison is done in E.coli only and with only two guides (Figure 5c). Among those two guides, one showed the same off-target profile (zero off-target rates) between known Cas12as and the tested variant. The other guide showed 7.3% off-target rates for AsCas12a and 0% for M44.

Response: We have modified the text at page 10 line 285-286 and line 291-298 to improve clarity as suggested.

Also, the on-target efficiency was compared to Cas9 (which cannot recognize T-rich PAMs

at all, Figure 3C, making this a rather unfair comparison). The more appropriate comparison would be to As/LbCas12as, which do recognize this PAM.

Response: Cas9 was used here as a negative control. The positive control is the WT MAD7 enzyme, which was one of the starting templates Cas12a enzymes and appeared in all of the identified and tested chimeras. We believe this is the right positive control to specifically address any changes in on- or off-target rates.

From Figure 6D, in mammalian system, the editing rates of M44 (~10%) is lower than wild-type MAD7 (~35%), and also appears to be lower than previously reported AsCpf1+engineered crRNA system (~50% across three different cell lines and multiple locus, Li et al., Nature Biomedical Engineering, 2017, PMID: 28840077). Although it is not a side-by-side comparison with the same targeting site, the reported editing rate is lower than the current state-of-the-art. Given that M44 on-target activity is 3-fold lower than wild-type MAD7, this seems to suggest that the chimeric strategy might not be able to yield better Cas variants. The specificity might be improved slightly here with the new variants, but this might just be because of the overall lower on-target efficiency.

Overall, I am unconvinced the M44 offers any real improvement. This negative result may merit publication but the presentation should be changed to emphasize this. Otherwise, the gene editing community may invest significant time and effort in an enzyme that is inferior to current Cas12 enzymes.

Response: We have modified the text at page 10 line 307-308, page 11 line 316-317, and page 12 line 352-353 to clarify that the point of this final demonstration was to prove that M44 retains function in mammalian system, and not to fully test the capabilities of this enzyme for editing in mammalian systems. In our experience there are many different ways to improve editing efficiency and there is extensive guide to guide variability, the combination of which would suggest that M44 has the potential to become an endonuclease worthy of additional investigation. We agree in full with the reviewer that on- and off- target efficiency are linked, and there are instances (e.g. allogenic cell therapies) where one might prioritize low off-target vs high on-target. The M44 chimera has significantly higher activity for some uncommon PAM sequences compared to AsCas12a and LbCas12a, potentially increasing the number of available target sites compared to commonly used nucleases. In such circumstances M44 might indeed prove superior. We have additionally modified the text as indicated above to make these points clear (page 11 line 338-340 and page 12 line 353-356).